# Genetic Characterization of a Neurovirulent West Nile Virus Variant Associated with a Fatal Great Grey Owl Infection

**DOI:** 10.3390/v13040699

**Published:** 2021-04-17

**Authors:** Katarína Peňazziová, Ľuboš Korytár, Patrik Pastorek, Juraj Pistl, Diana Rusňáková, Tomáš Szemes, Viktória Čabanová, Martina Ličková, Kristína Boršová, Boris Klempa, Tomáš Csank

**Affiliations:** 1Department of Microbiology and Immunology, University of Veterinary Medicine and Pharmacy in Košice, Komenského 73, 04181 Košice, Slovakia; k.penazziova@gmail.com (K.P.); juraj.pistl@uvlf.sk (J.P.); 2Department of Epizootiology, Parasitology and Protection of One Health, University of Veterinary Medicine and Pharmacy in Košice, Komenského 73, 04181 Košice, Slovakia; lubos.korytar@uvlf.sk; 3ZOO Košice, Ulica k Zoologickej Záhrade 1, 04001 Košice, Slovakia; pastorek@zookosice.sk; 4Department of Molecular Biology, Faculty of Natural Sciences and Science Park, Comenius University in Bratislava, Ilkovičova 8, 84104 Bratislava, Slovakia; diana.rusnakova@uniba.sk (D.R.); tomas.szemes@uniba.sk (T.S.); 5Biomedical Research Center of Slovak Academy of Sciences, Department of Virus Ecology, Institute of Virology, Dúbravská Cesta 9, 84505 Bratislava, Slovakia; viktoria.cabanova@savba.sk (V.Č.); virulic@savba.sk (M.L.); kristina.borsova@savba.sk (K.B.); boris.klempa@savba.sk (B.K.); 6Department of Microbiology and Virology, Faculty of Natural Sciences, Comenius University in Bratislava, Ilkovičova 6, 84215 Bratislava, Slovakia

**Keywords:** West Nile virus, great grey owl, virulence determinant, zoological garden, zoonosis

## Abstract

This study reports on a fatal case of a captive great grey owl infected with the West Nile virus (WNV) in the zoological garden Košice, eastern Slovakia (Central Europe). The tissue samples of the dead owl were used for virus isolation and genetic characterization. The novel isolate is genetically closer to Hungarian, Greek, and Bulgarian strains from the central/southern European clade of lineage 2 than to the strains previously isolated in Slovakia. Interestingly, it carries NS3-249P, a molecular virulence determinant associated with higher neurovirulence, which has not previously been observed in Slovakia. Subsequent serological investigation of the captive owls revealed additional seropositive animals, indicating local WNV transmission. Although no WNV-positive mosquitoes were found, the presence of the WNV principal vector *Culex pipiens* complex together with the described fatal case and further serological findings indicate an endemic focus of bird-neurovirulent WNV variant in the area.

## 1. Introduction

West Nile virus (WNV) and Usutu virus (USUV), both from genus *Flavivirus* (*Flaviviridae*), are included in the antigenic complex of the Japanese encephalitis virus (JEV). The natural transmission cycle of all members of the JEV antigen complex includes birds as the main amplification hosts and several mosquito species as vectors. The mosquito becomes infected during blood-sucking on a viremic bird and remains persistently infected throughout its life, potentially transmitting the virus to all vertebrates on which it feeds. Horses and humans are considered dead-end hosts [1].

WNV was isolated for the first time from the blood sample of a woman in the West Nile district in Uganda in 1937 [2]. Thereafter, WNV became one of the most widespread arboviruses distributed in Africa, Asia, Europe, and Australia. In 1999, the virus reached North and later South America and caused epidemics and epizootics on both continents [3].

In humans, WNV infection can lead to a severe zoonotic disease. The course of the infection is in most cases asymptomatic, but in approximately 20%, it has flu-like symptomatology. The neuroinvasive disease develops in less than 1% of infections. However, in those cases, the fatality may reach 20% and can cause chronic illness in survivors [4].

In animals, WNV infection is usually asymptomatic, although it may cause fever, and some horses may experience encephalomyelitis, among which approximately 30% die or approximately 10–20% recover with a neurological sequel [5]. In birds, WNV may cause fatal systemic disease complicated by encephalitis, with different sensitivities among avian families. Corvids, owls, and birds of prey (Corvidae, Accipitridae, Falconidae, and Strigidae) are particularly susceptible to infection [6].

In Slovakia, WNV was first isolated from *Aedes cantans* mosquitoes between the years 1960 and 1972 [7]. After that, serological studies showed WNV circulation in horse and bird populations [8,9,10,11]. The phylogenetic analysis of whole-genome strains isolated from the brain of dead raptors showed relatedness to WNV lineage 2 [12]. The circulation of WNV lineage 2 was also confirmed in mosquitoes by Čabanová et al. (2019). Viral RNA was detected in *Cx. modestus* and *Cx. pipiens* mosquitoes collected in the Komárno district. BLAST analysis of a partial NS5 gene showed almost 100% identity with strains from Germany, Italy, and Austria [13]. The first confirmed autochthonous case of West Nile fever was diagnosed in the late summer of 2019, in a man infected by mosquitoes in the area of western Slovakia [11,14,15].

This study was prompted by a case of a dead great grey owl (*Strix nebulosa lapponica*) kept in the zoological garden (zoo) in Košice (Eastern Slovakia). Tissue samples were analyzed by a Pan-Flavi molecular approach, virus isolation, and genetic characterization. In addition, serological examination of the captive owls and screening of WNV in captured mosquitoes in the zoo and city of Košice was performed to further evaluate the risk of the local mosquito-borne flavivirus transmission.

## 2. Materials and Methods

### 2.1. Description of the Study Area—Zoo Košice

Zoo Košice is located in the Čierna Hora Mountains, 453 m above sea level (coordinates: 48°47′10″ N 21°12′03″ E) in eastern Slovakia. It is the largest zoo in Central Europe, covering an area of 289 hectares (ha) in the neighborhood of Kavečany village near Košice city. The area of Zoo Košice includes habitats of both urban and rural types. The owls in Zoo Košice are kept in outdoor cages. Constructions of cages are not secured against the entering of vectors such as mosquitoes and potential reservoirs of WNV and USUV, such as house sparrows (*Passer domesticus*) and tree sparrows (*Passer montanus*). Both sparrow species roost and forage in cages throughout the year, and close contact between owls, mosquitoes, and passerines is maintained. In addition, a large population of corvids, including Eurasian magpies (*Pica pica*), common ravens (*Corvus corax*), and hooded crows (*Corvus cornix*), live in the area of the zoo.

### 2.2. Flavivirus RNA Detection in Brain Tissue

Nucleic acids were isolated from the brain of the dead owl by QIAamp^®^ cador^®^ Pathogen Mini Kit (Qiagen, Hilden, Germany). The cDNA was synthesized by LunaScript RT SuperMix (New England Biolabs, Ipswich, MA, USA). The presence of flavivirus RNA was examined using a semi-nested PCR (Quick-Load^®^ Taq 2 × Master Mix, New England Biolabs, Ipswich, MA, USA) by amplification of a partial non-structural (NS) 5 gene. In the first run, PanFlavi-F and cFD2 primers were used to amplify a 599 bp fragment, and in the second run, the primer pair of PanFlavi-F and PanFlavi-R amplified a 360 bp fragment [16,17]. Amplicons were extracted from an agarose gel (Wizard gel, Promega, Madison, WI, USA) and analyzed by Sanger sequencing (Microsynth, Balgach, Switzerland).

### 2.3. Virus Isolation and Virus Quantification

For virus isolation and quantification, 10% (*w/v*) tissue suspensions were prepared from the owl brain, lung, heart, liver, and kidney in 10% Eagle’s minimum essential medium (EMEM) supplemented by 1% of penicillin G sodium salt (0.06 g/L) and streptomycin (0.1 g/L, Biowest, Nuaillé, France); 0.5% amphotericin B (250 mg/mL, Pan Biotech, Aidenbach, Germany) and 0.1% gentamicin sulfate (50 mg/mL, Biowest; 10% EMEM-ATB). Samples were homogenized with ten 2.8 mm stainless beads in Precellys 24 Dual (Bertin Technologies, Aix-en-Provence, France) at 4000 rpm for 3 × 10 s with a pause at 15 s. After centrifugation at 3800× *g* for 20 min at 4 °C, the tubes were moved onto ice, and suspensions were then incubated for at least 20 min to let the aerosol settle down. 

For virus isolation, 0.5 mL of 10 % brain suspension was used to infect Vero E6 sub-confluent monolayer cultivated on a T75 flask (Greiner Bio-One, Frickenhausen, Germany). For virus quantification, 100 µL of the 10% suspension of each tissue was 10-fold diluted in 10% EMEM-ATB up to 10^−6^. Monolayers of cell line Vero E6 were infected with 250 μL of each dilution in duplicates on 12-well plates. After 1 h adsorption at 37 °C, the cells were washed and overlaid with 2% carboxymethylcellulose in 1% EMEM-ATB. Four days after infection, the cells were fixed with 10 % buffered formalin, washed with phosphate-buffered saline (PBS), and stained with 0.1 % crystal violet. The virus load was expressed as plaque-forming units per milliliter (PFU/mL) of tissue suspension.

### 2.4. Sequencing of the WNV Isolate

To obtain a complete genomic sequence of WNV, a next-generation sequencing approach was used. Total RNA was purified from the cell culture medium of infected cells using the QIAamp Viral RNA Minikit (Qiagen, Hilden, Germany). Obtained RNA was then subjected to random-primed reverse transcription PCR (RT-PCR) as described in Reference [18]. Generated RT-PCR products were purified using the Clean & Concentrator kit (Zymo, Freiburg, Germany). Sequencing libraries were prepared using the Nextera Library Prep Kit (Illumina, San Diego, CA, USA) according to the standard protocol. Prepared libraries were sequenced with the Miseq reagent kit V2 (Illumina, San Diego, CA, USA) using paired-end sequencing with a 120 bp read length. The obtained sequences were analyzed and mapped to the reference sequence (MH244511) with Geneious software (Biomatters, Auckland, New Zealand). Gaps in the genome after NGS were sequenced using primers published by Reference [19]. The nucleotide sequence of the WNV 769.B/2018/Kavečany/SVK isolate was deposited in GenBank with accession numbers MW561633.

### 2.5. Phylogenetic Analyses

Molecular phylogenetic analysis of polyprotein of WNV isolated from the owl was performed with the maximum likelihood method, and the choice of the substitution method was made by the Find Best DNA/Protein Models in MEGA X [20]. The evolutionary history was inferred using the maximum likelihood method and Jones-Taylor-Thornton (JTT) matrix-based model [21]. The percentage of trees in which the associated taxa clustered together is shown next to the branches. Initial trees for the heuristic search were obtained automatically by applying Neighbor-Join and BioNJ algorithms to a matrix of pairwise distances estimated using the JTT model and then selecting the topology with a superior log-likelihood value. Evolutionary analyses were conducted in MEGA X.

### 2.6. Serological Surveillance of Captive Owls in Zoo Košice

All sampled individuals were clinically healthy. Blood samples were obtained by puncture of the right jugular vein according to [22]. Seven blood samples were collected from snowy owls (*Bubo scandiacus*; *n* = 2), great grey owls (*n* = 2), northern hawk-owl (*Surnia ulula*; *n* = 1), and Eurasian eagle-owls (*Bubo*; *n* = 2) in February 2019.

Owl serum samples were serologically tested for WNV total antibodies using a blocking ELISA kit (INGEZIM WNV COMPAC, Ingenasa, Madrid, Spain) following the manufacturer’s protocol without modifications. Results are expressed as the inhibition percentage (IP), where IP ≥ 40% is considered positive, IP ≤ 30% is considered negative, and IP between those two values is considered doubtful. According to the manual, doubtful results should be confirmed by the seroneutralization assay.

There is no commercial USUV ELISA kit for birds; hence, the samples were further tested by simultaneous microtitration neutralization tests for USUV and WNV antibody detection. A total of 25 mL of 1:5 pre-diluted samples was heat-inactivated (56 °C/30 min). Serial dilution up to 1:320 was conducted in duplicate wells, and equal volumes of both viruses were added into each well. We used the WNV isolate 291.B/2013/Velky Biel/SVK (MH244511) once passaged and the USUV strain 939/01 (kindly provided by Norbert Nowotny, Veterinary University Vienna) four times passaged, both on Vero E6 cells. After incubation at 37 °C for one hour, 50 μL of Vero E6 cells were added into each well. Both inoculums were back-titrated in triplicates and expressed as mean median tissue culture infectious dose (TCID_50_) in 25 μL. We observed 4.3 × 10^1^ and 1.6 × 10^2^ for WNV and USUV, respectively.

### 2.7. Mosquito Screening in Zoo Košice

In the zoo, six collections were carried out between the 20th of June to 13th of September 2019 in four sites located in the accessible area (approximately 75 ha). One trap was placed directly behind the aviary of the dead owl; three other traps were placed on the banks of lakes in the zoo. Adult females were collected by traps (BG-Sentinel 2, Biogents AG, Regensburg, Germany) using CO_2_. Captured mosquitoes were transported to the laboratory as soon as possible and stored at −80 °C. Morphological identification was carried out on a Petri dish placed on ice under a Stereomicroscope SZO-4 (Optika, Ponteranica, Italy) according to the morphological key [23].

### 2.8. Processing of Mosquitoes for WNV Detection

Mosquitoes were pooled based on the species and date and site of the collection with a maximum number of 25 individuals per pool. Homogenization was carried in the same way as that for the tissue (described above) with three stainless beads at 2500 × rpm. The suspensions were clarified at 4 °C for 10 min (5000× *g*). A total of 150 μL of the supernatant was immediately used for RNA extraction.

The viral RNA was extracted using the NucleoSpin^®^ RNA Virus kit (Macherey Nagel, Düren, Germany) according to the “Viral RNA isolation from cell-free biological fluids”, and RNA was reverse-transcribed as described above. Reverse transcription real-time PCR (RT-qPCR) was performed using the Luna^®^ Universal Probe qPCR Master Mix (New England Biolabs, Ipswich, MA, USA) following the protocols [24].

## 3. Results

### 3.1. Detection and Quantification of WNV from the Dead Owl

A 17-years old great grey owl died on 29 August 2018. The care workers observed apathy and inability to perch one day before death. The owl lived in Zoo Košice for two years with no travel history. Brain, lung, heart, liver, and kidney samples were collected and used for further examination.

The brain tissue was used for initial identification of the causative agent by flavivirus generic RT-PCR and virus isolation. Sequence analysis of the obtained PCR product showed the presence of WNV lineage 2. Tissue suspension used for virus isolation caused almost 100% cytopathic effect (CPE) 5 days after infection. The isolate was designated as WNV 769.B/2018/Kavečany/SVK and was deposited in the European Virus Archive.

The highest virus loads were observed in the brain (5.45 × 10^7^ PFU/mL) and heart (1 × 10^6^ PFU/mL), followed by several log lower PFUs in the lungs (6.2 × 10^3^ PFU/mL), liver (1.8 × 10^3^ PFU/mL), and kidney (8 × 10^2^ PFU/mL).

### 3.2. Genetic Characterization

The complete genome of the WNV 769.B/2018/Kavečany/SVK isolated in cell culture was determined. The genome consists of 10,964 nt and contains one open reading frame located between 94 and 10,398 nt, which encodes a 3434 aa long polyprotein.

Phylogenetic analysis of the complete polyprotein amino acid sequences showed that the new Slovak WNV isolate belongs to the central/southern clade of lineage 2 (Figure 1). Within the central/southern clade, the WNV 769.B/2018/Kavečany/SVK was grouped on the same branch with the Hungarian 578/10 strain (KC496015) isolated from the brain of a horse. These two isolates shared a common ancestor with the Greek (HQ537483, KJ577738-39, KJ883343-46, and KJ883348-49) strains from the central Macedonian and east Macedonian/Thracian clusters and the Bulgarian (KU206781) strain. Amino acid sequence comparison of the new isolate showed common amino acid residues in the non-structural proteins in several positions, namely, isoleucine at NS2B-119 (NS2B-119I), proline at NS3-249 (NS3-249P), glycine at NS4B-14 (NS4B-14G), and alanine at NS4B-49 (NS4B-49A). These residues are characteristic of the Greek cluster (Table 1). Methionine at NS4B-113 (NS4B-113M) is also characteristic of the Greek cluster but not present in the new Slovak and Hungarian 578/10 strains. These strains possess valine (NS4B-113V), which was also present in the rest of the strains included in the comparison (Table 1).

A lysine was present in the new isolate at position NS1-44 (NS1-44K; Table 1), which was also found in strains from Romania (KJ934710), Ukraine (JX041631), and South Africa (EF429198 and EF429197). In the other strains, an arginine at position NS1-44 (NS1-44R) was observed. The new isolate has two unique amino acids in positions E-399 and NS3-486, namely an arginine (E-399R) and a leucine (NS3-486L; Table 1).

The tree with the highest log likelihood (−11087.44) is shown. A discrete gamma distribution was used to model evolutionary rate differences among sites (five categories (+G, parameter = 0.0500)). The tree is drawn to scale, with branch lengths measured in the number of substitutions per site. This analysis involved 33 amino acid sequences, with 3434 positions in the final dataset.

### 3.3. Serological Examination of Owls

Two samples (*n* = 7) tested positive for WNV antibodies in ELISA. The presence of antibodies against WNV was determined in one male Eurasian eagle-owl (IP = 89.9%), which was hatched in the Czech Republic on 23th of May 1989 and lived in Zoo Košice since 21 November 1989. The second seropositive bird was a great grey owl male (IP = 65.6%) which was imported from Hodonín, the Czech Republic, to Zoo Košice on 10 October 2018.

Due to the low volume of serum samples, only the eagle-owl’s serum was tested by microtitration neutralization test. It showed neutralizing activity against WNV with a titer of 1:320.

### 3.4. Mosquito Screening in Zoo Košice 

Overall, 119 female mosquitoes in 16 pools were tested for the presence of WNV by RT-qPCR. The vast majority of mosquitoes belonged to the *Culex pipiens* complex (*n* = 115). Other species were *Aedes caspius*, *Ae. vexans*, *Ae. sticticus,* and *Anopheles plumbeus*, each represented by one individual. None of the tested pools was positive for WNV.

## 4. Discussion

In general, zoological gardens are ideal epidemiological and epizootiological monitoring stations. Despite the high mortality rate of American crows (*Corvus brachyrhynchos*) caused by WNV in 1999 in North America, attention was not paid to the infection until a few dead birds were found in the Bronx Zoo/Wildlife Conservation Park. The mortality of birds with clinical signs increased to 70% [26].

Owls are suitable sentinels in the serological and virological surveillance of WNV and USUV because several species are highly susceptible to the infection. They may exhibit a wide range of clinical signs, and specific lesions in naturally infected owls (Strigidae) can be observed in almost all internal organs [27,28,29,30,31]. Ziegler et al. (2019) described fatal WNV cases in two great grey owls, two northern goshawks, and two common blackbirds in zoological gardens in Germany. The owls suddenly died without any clinical signs. The virus load measured by real-time RT-PCR in the brain, kidney, spleen, and heart reached a high cycle threshold value (C_t_ = 14–20) [32]. Higher susceptibility of owls to the WNV infection was also evident during an outbreak in the Owl Foundation, Canada in September 2002, when 44% (108/245) of the owls died and the WNV-related death rate for the great grey owls was 91.3% [33].

A circulation of WNV in endemic areas may result in a decline in many avian species, including owls, as some owl species suffer greater than 90% death rates due to WNV [33,34,35]. In the context of susceptibility and high mortality of the great grey owl to WNV [28], the spread of a potentially highly virulent strain into higher latitudes may have an impact on the population of this species. The isolated WNV from the brain of a dead great grey owl in Zoo Košice (WNV 769.B/18/Kavečany/SVK) is genetically closer to the Hungarian, Greek, and Bulgarian strains than to strains previously isolated in Slovakia. Each of them, except the Nea Santa-Greece-2010 strain, was isolated or detected in human or equine cases of West Nile neuroinvasive disease [36,37,38,39].

A higher neurovirulence of the identified strain is also suggested by the presence of the proline at NS3-249, previously associated with higher neurovirulence [39]. NS3 is a multifunctional protein consisting of domains with protease, helicase, nucleoside triphosphatase, and RNA triphosphatase activities. These activities are believed to be vital in the flavivirus replication cycle, virion budding, and membrane reorganization. NS3 is also known to be associated with apoptosis induction and has been suggested to play a role in neurovirulence [40]. Brault et al. (2007) studied the influence of NS3-249P on avian virulence in the American crow model [39]. Parental lineage 1 strains (the North American NY99 strain 382-99 and the Kenyan KN-3829 strain) differed by a single amino acid within the NS3-249 helicase: threonine in KN-3829 and proline in NY99. The introduction of threonine to the virulent WNV NY99 lineage 1 strain significantly decreased bird mortality (from 100 to 12.5%), while the introduction of proline in the low-virulence KN-3829 strain increased bird mortality from 31 to 94%. In addition, in silico structural modeling showed that NS3-249P could cause rotational constraints on the flexibility of the helicase loop. WNV NS3 protein mutants with alanine, threonine, and proline were tested in vitro for ATPase activity at different temperatures. Each of them had similar kinetic profiles at 28 and 37 °C, but only NS3-249P retained its activity at 42 °C, which represents avian body temperature. Results suggest that NS3-249P probably contributes to increased protein stability under higher temperatures [39]. Szentpali-Gavallér et al. (2016) studied the influence of proline on histidine substitution at NS3-249 on Vero E6 cells, which significantly lowered the multiplication kinetics 48 h after infection [36].

Some owls serologically examined in our study had spent many years in Zoo Košice, while others had a recent travel history. Serological evidence of the WNV-specific antibodies in 30-year-old male Eurasian eagle-owl without travel history for several decades indicates an autochthonous infection. A second seropositive male great grey owl was transferred from the Czech Republic to Zoo Košice 4 months before testing. Hence, previous exposure to WNV in the place of origin cannot be excluded.

The circulation of WNV among the vector population in Zoo Košice was not confirmed during our surveillance, and only a limited number of mosquitoes were collected. However, the presence of a principal vector of WNV in Slovakia—the *Cx. pipiens* complex [13]—in the proximity to the birdcages suggest a potential risk for further spreading of a novel, more virulent WNV variant within the zoo and also to surrounding areas.

## 5. Conclusions

The isolation of WNV from the brain of a dead great grey owl in Zoo Košice uncovered the presence of a novel WNV lineage 2 variants in Slovakia. The novel isolate is genetically closer to the Hungarian, Greek, and Bulgarian strains from the central/southern clade than to WNV strains previously detected in Slovakia. According to the proline at position NS3-249 found in the new isolate, it is assumed that this variant may have a higher avian virulence. WNV antibodies were also detected in two other owls in Zoo Košice. Although no WNV RNA was found in mosquito pools, the suitable vector occurrence and the WNV circulation among reservoirs suggest the spread of a new and more virulent variant of WNV in the area.

## Figures and Tables

**Figure 1 viruses-13-00699-f001:**
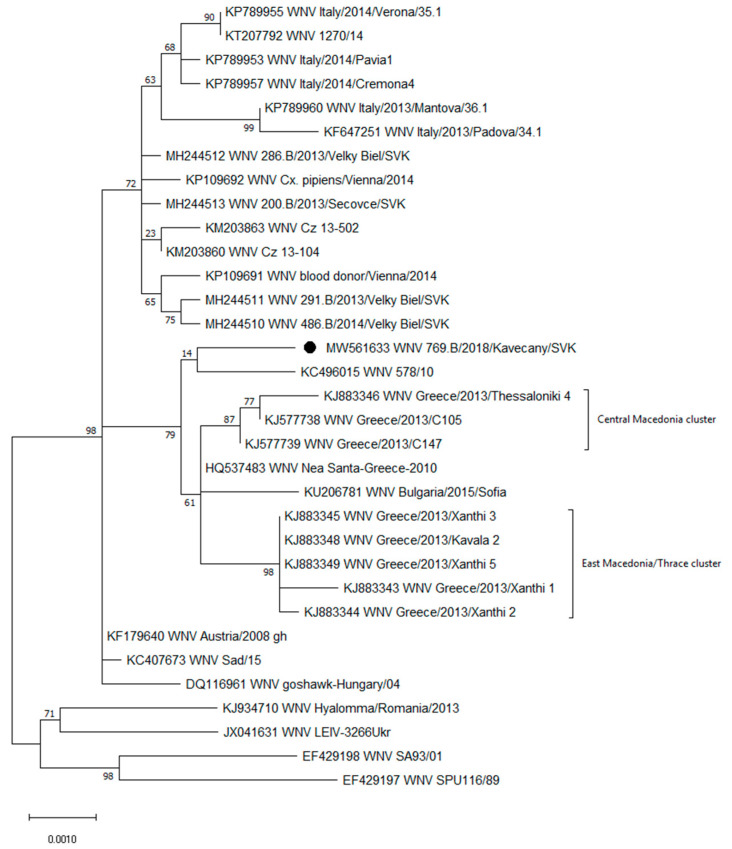
The maximum likelihood phylogenetic tree based on West Nile virus (WNV) lineage 2 complete polyprotein amino acid sequences indicating the phylogenetic placement of the new cell culture isolate 769.B/2018/Kavečany/SVK, which is marked with a point. Clusters are labeled according to [25].

**Table 1 viruses-13-00699-t001:** Comparison and position of substitutions in the polyprotein of WNV lineage 2 strains.

Accession Number	E		NS1	NS2B	NS3		NS4B			NS5
	159T (449)	399R (689)	44K (835)	119I (1493)	249P (1754)	486L (1991)	14G (2287)	49A (2322)	113V (2386)	298T (2827)
MW561633^■^/SVK	T	R	K	I	P	L	G	A	V	T
KC496015^□^/HUN	I	K	R	.	.	F	.	.	.	A
KU206781^▼^/BGR	M	K	R	.	.	F	.	.	M	A
HQ537483^●^/GRC	I	K	R	.	.	F	.	.	M	A
KJ883349^▼^/GRC	I	K	R	.	.	F	.	.	M	A
KJ883345^▼^/GRC	I	K	R	.	.	F	.	.	M	A
KJ883348^▼^/GRC	I	K	R	.	.	F	.	.	M	A
KJ883343^▼^/GRC	I	K	R	.	.	F	.	.	M	A
KJ883344^▼^/GRC	I	K	R	.	.	F	.	.	M	A
KJ883346^▼^/GRC	I	K	R	.	.	F	.	.	M	A
KJ577738^▼^/GRC	I	K	R	.	.	F	.	.	M	A
KJ577739^▼^/GRC	I	K	R	.	.	F	.	.	M	A
MH244513^■^/SVK	.	K	R	V	H	F	S	T	.	.
MH244512^■^/SVK	.	K	R	V	H	F	S	T	.	.
MH244511^■^/SVK	A	K	R	V	H	F	S	T	.	.
MH244510^■^/SVK	A	K	R	V	H	F	S	T	.	.
KM203863^●^/CZE	.	K	R	V	H	F	S	T	.	.
KM203860^●^/CZE	.	K	R	V	H	F	S	T	.	.
KP789955^▼^/ITA	.	K	R	V	H	F	S	T	.	.
KP789957^▼^/ITA	.	K	R	V	H	F	S	T	.	.
KP789953^▼^/ITA	.	K	R	V	H	F	S	T	.	.
KP789960^▼^/ITA	.	K	R	V	H	F	S	T	.	.
KF647251^▼^/ITA	.	K	R	V	H	F	S	T	.	.
KT207792^●^/ITA	.	K	R	V	H	F	S	T	.	.
DQ116961^■^/HUN	I	K	R	V	H	F	S	T	.	A
KP109692^●^/AUT	.	K	R	V	H	F	S	T	.	.
KP109691^▼^/AUT	A	K	R	V	H	F	S	T	.	.
KF179640^■^/AUT	I	K	R	V	H	F	S	T	.	A
KC407673^■^/SRB	I	K	R	V	H	F	S	T	.	A
KJ934710^○^/ROU	M	K	.	V	H	F	S	T	.	A
JX041631^■^/UKR	I	K	.	V	H	F	.	T	.	A
EF429198^▼^/SAF	I	K	.	V	H	F	S	T	.	A
EF429197^▼^/SAF	I	K	.	V	H	F	S	T	.	A

Legend: position of aa in the polyprotein is in parentheses; the shapes indicate hosts of isolation or detection of certain strains: ^■^ bird, ^□^ horse, ^▼^ human, ^●^ mosquito, ^○^ tick. The names of strains used in the polyprotein comparison are depicted in the phylogenetic tree in Figure 1. Country codes: SVK—Slovakia, HUN—Hungary, BGR—Bulgaria, GRC—Greece, CZE—Czech Republic, ITA—Italy, AUT—Austria, SRB—Serbia, ROU—Romania, UKR—Ukraine, SAF—South Africa.

## Data Availability

The data that support the findings of this study are available from the corresponding author upon reasonable request.

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
