# Peer review of "Genetic Characterization of a Neurovirulent West Nile Virus Variant Associated with a Fatal Great Grey Owl Infection"

_viruses, 2021, doi:10.3390/v13040699_

Round 1

Reviewer 1 Report

In the paper “Genetic Characterization of a Neurovirulent West Nile Virus Variant Associated with a Fatal Great Grey Owl Infection” the authors describe the identification of a novel isolate of West Nile virus, that they isolated from a dead owl in a zoo in Slovakia. The paper is easy to read and very clear. The importance of the work and the significance of finding this new isolate comes across clearly when reading through the paper. The authors have taken a very straight forward approach and I don’t have major comments about the authors works.  

My minor comments are –

Line 98, can it be made a little clearer how the 10% suspensions were made. It just reads a bit off as it starts with “ten percent suspensions were prepared ….. in 10% EMEM”. Also, since this is a Virus Isolation section, the authors haven’t actually said how they isolated the virus. The section mainly talks about tissue homogenization and plaque assays.

Line 174, is that 150 ml or ul?

Line 218, is localization the best word here? It’s just the position where substitutions are seen that is reported.

Line 219, is it nucleotide position that is in parentheses? needs specifying. The new isolate isn’t highlighted in bold in the table.

Line 220, is that meant to say row 1 instead of figure?

Line 264, “when 44% (108/245) owl died”, should say when 44% (108/245) of the owls died.

Line 310, strain. Is it a strain or an isolate? the paper talks about an isolate, and now it's called a strain. Just need to be consistent.

Reviewer 2 Report

The presented manuscript is well-written, with appropriate study design and  methodology, clearly presented results and valuable discussion. Please see below some minor comments.

Introduction

Line 47: please correct severe zoonotic disease (instead of serious zoonotic disease)

Line 49: please correct neuroinvasive disease (instead of neurological disease)

Line 55: Please correct Corvids (Uppercase letter)

Results

Line 235: Please correct  "  ... tested positive for WNV antibodies in ELISA. "

Discussion

Line 261: Please correct real-time RT-PCR

Line 274: Please correct WNV neuroinvasive disease.
